# Pandemic Events Caused by Bacteria Throughout Human History and the Risks of Antimicrobial Resistance Today

**DOI:** 10.3390/microorganisms13020457

**Published:** 2025-02-19

**Authors:** Pedro Filho Noronha Souza, Nicholas Silva dos Santos Filho, João Lucas Timbó Mororó, Daiane Maria da Silva Brito, Ana Beatriz da Lima, Felipe Pantoja Mesquita, Raquel Carvalho Montenegro

**Affiliations:** 1Laboratory of Bioinformatics Applied to Health, Drug Research and Development Center (NPDM), Federal University of Ceará, Fortaleza 60430-275, CE, Brazil; nicholassantos@alu.ufc.br (N.S.d.S.F.); jlmororo2005@gmail.com (J.L.T.M.); daianebmaria@outlook.com (D.M.d.S.B.); anabeatrizdalima@gmail.com (A.B.d.L.); felipemesquita05@gmail.com (F.P.M.); rcm.montenegro@gmail.com (R.C.M.); 2Drug Research and Development Center, Department of Physiology and Pharmacology, Federal University of Ceará, Fortaleza 60430-275, CE, Brazil; 3National Institute of Science and Technology in Human Pathogenic Fungi (FunVir), Faculty of Pharmaceutical Sciences of Ribeirão Preto, University of São Paulo, Ribeirão Preto 14040-903, SP, Brazil; 4Researcher at the Cearense Foundation to Support Scientific and Technological Development, Fortaleza 60325-452, CE, Brazil

**Keywords:** antimicrobial resistance, bacterial resistance, antimicrobial peptides

## Abstract

During human history, many pandemic events have threatened and taken many human lives over the years. The deadliest outbreaks were caused by bacteria such as *Yersinia pestis*. Nowadays, antimicrobial resistance (AMR) in bacteria is a huge problem for the public worldwide, threatening and taking many lives each year. The present work aimed to gather current evidence published in scientific literature that addresses AMR risks. A literature review was conducted using the following descriptors: antimicrobial resistance, AMR, bacteria, and Boolean operators. The results showed that antimicrobial-resistant genes and antibiotic-resistant bacteria in organisms cause critical infectious diseases and are responsible for the infections caused by antibiotic-resistant bacteria (ARB). This review emphasizes the importance of this topic. It sheds light on the risk of reemerging infections and their relationship with AMR. In addition, it discusses the mechanisms and actions of antibiotics and the mechanisms behind the development of resistance by bacteria, focusing on demonstrating the importance of the search for new drugs, for which research involving peptides is fundamental.

## 1. Introduction

Antimicrobial resistance in bacteria (AMRB) is a process that could be inhibited. It is part of natural cellular evolution. However, this process has many problems for human and animal health, as the number of infections caused by AMRB grows yearly. Infections caused by AMRB require an investment of 2 to 3.5% of the gross domestic product to address [1]. It is estimated that more than 10 million deaths will occur annually [2]. Although many bacteria are harmless under normal circumstances, infections caused by AMRB are challenging to treat and often require a multi-drug approach, leading to severe disease and death [3,4]. Given the fast development of resistance, it is hard to detect AMRB infections and rapidly start a treatment strategy to prevent the re-emergence of bacteria in their native habitat that have been trapped in permafrost for thousands of years.

The misuse of antibiotics has significantly influenced the historical development of bacterial resistance, both in the past and today. Since the discovery of antibiotics like penicillin, bacteria have rapidly evolved mechanisms to resist these drugs. This resistance is accelerated by overprescribing antibiotics for viral infections, patients not completing their full course of treatment, using antibiotics in livestock to promote growth, and self-medication without the proper guidance [1,3,5]. These actions expose bacteria to antibiotics unnecessarily, allowing resistant strains to thrive and multiply. Consequently, infections caused by resistant bacteria have become more challenging to treat, leading to increased mortality, prolonged illness, and higher medical costs. The misuse of antibiotics underscores the urgent need for responsible antibiotic use and robust stewardship programs to mitigate the growing threat of antibiotic resistance.

Over the years, bacteria have developed many mechanisms of resistance (Figure 1) that allow them to become immune to the effect of antibiotics [5,6]. This resistance is a natural process of cellular evolution. However, the misuse of antibiotics in association with disinformation in the application of antibiotics in clinics has accelerated the process in a way that was not expected [2].

According to a systematic review in 2019, an estimated 4.95 million new deaths were caused by AMRB in 2019. For example, the World Health Organization (WHO) reported that some regions of the world are most affected by AMRB, such as Australasia, at 6.5 deaths per 100,000, and sub-Saharan Africa, at 27.3 deaths per 100,000 [7,8].

In 2019, AMRB was responsible for the highest numbers of three infections: (1) bloodstream infections, lower respiratory and thorax infections, and intra-abdominal infections, which represented 79% of deaths caused by AMRB, as reported by the WHO [8]. Based on that, this study aimed to explore the problem related to AMR in bacteria, discussing the consequences of this problem during human history and today and advances in this area [7,8].

The last global burden analysis on antimicrobial resistance of bacteria, published in September 2024, analyzed the data from 1990 to 2021 and predictions to 2050. The results of these analyses forecast an estimation of 8.22 million deaths associated with AMRB [9]. Additionally, AMR has an economic impact on human and animal health, leading to annual treatment costs of USD 1 trillion to USD 3.4 trillion by 2050 [10]. Based on that, this study aims to examine the history of pandemics caused by bacteria and how the resistance could lead to a new outbreak caused by resistant bacteria.

This review discussed the pandemic events during human history caused by *Yersinia pestis* to remember how harmful and lethal bacteria can be and thus warn about the problem of emergent bacterial infections caused by resistant bacteria that might lead to new pandemic events.

## 2. Methods

A literature review addressed the following question: “Does antimicrobial resistance increase the resurgence of pathogens that threaten human health?” The inclusion criteria were as follows: (1) studies published in the English language; (2) articles indexed by the electronic databases Medical Literature Analysis on MEDLINE and PUBMED, the most important repository of medical studies; and (3) articles published in journals that possess a rigorous peer-review process. The publications in the last 5 years were individually searched and selected by investigators in November 2022. The studies that did not meet these criteria were automatically excluded. More than 200 articles were found. However, after the criteria, 66 were used for this study.

The Preferred Reporting Items for Systematic Review and Meta-Analyses (PRISMA) guidelines [11] were followed to select the studies. A search strategy was created by choosing the words *bacteria*, *bacterial resistance*, *bacterial outbreaks*, and *infections caused by resistant bacteria*. The search used filters such as years, free full text, meta-analysis, review, systematic review, English, and Medline.

## 3. Results and Discussion

### 3.1. Historical Sources

Throughout human history, several outbreaks caused by bacteria have taken many lives and threatened many more (Figure 2). Three historical epidemics referred to as “plague” (Figure 2) threatened humans by devastating the population at the time. The situation worsened with the emergence of the first event of bacterial resistance (Figure 1), and hence with the discovery that the thawing of the permafrost could bring to light bacteria that humans know nothing about [12].

#### 3.1.1. The Plague of Justinian (541 to 750/767)

This outbreak, which was caused by *Yersinia pestis* and ran from 541 to 750, was called the black death and killed almost 100 million people. It was also named the “plague of Justinian”, based on Justinian I, the emperor of the Roman Empire of the Orient (Figure 2). The outbreak had 14 to 21 waves, leading to millions of deaths. It is estimated that half of the European population died from the black death [13,14]. According to Evagrius Scholasticus, the plague originated in Ethiopia [14,15]. However, other hypotheses for the first case have been proposed, and the real information is still unknown.

The outbreak started as an endemic disease. However, throughout the waves, it spread to large areas and reached other continents, such as Europe. The plague spread deeply into the inland areas via human movements and along trade routes until it reached Germany, England, and Ireland [16].

The symptoms presented by infected people were high fever, buboes, intense headache, and rapid death. That information provided a clinical clue about the etiological agent of the plague. The historian Procopius described these epidemics as follows: “It generally happened that those whose buboes grew large and suppurated, recovered from the disease, which seemed to spend its violence upon these tumors; while in those whose buboes remained without suppuration, it had an unfavorable termination”. According to Gregory of Tours, death was sudden [16].

There are still many controversies about the plague. The first is about the number of deaths. On one hand, a group of authors estimated deaths of about 15 to 100 million victims, which is equivalent to 60% of the entire human population at that time [17,18,19]. By contrast, one recent historical and archaeological study suggested several deaths totaling 0.1% of the estimated human population. Therefore, this discussion proposed that the Justinian Plague had no substantial impact on human life, as described before [13].

#### 3.1.2. Second Pandemic (1346 to the 18th Century)

The Florentine poet Petrarch set the stage for the plague’s return in the mid-14th century [20]. The second pandemic likely began in the late 1330s in Central Asia, probably in what is now Kazakhstan, Russia, or China [16]. After the alleged first instance of bacteriological warfare in 1346 [21,22], the plague spread through sea routes from Caffa to Constantinople, North Africa, and Western Europe.

The spread of the deadly disease was rapid and relentless. As noted by Signolli [12] and [23], merely speaking to the afflicted was enough to contract the fatal illness, leaving no hope for recovery. Initially, the path of the disease and the chronology of infected cities were easily traceable. However, as the plague began to follow travelers and their goods, its spread became more unpredictable, stopping at various waypoints along trade routes. During the first seven years of the second pandemic, from 1346 to 1353, this first wave of the plague—historically known as the Black Death—devastated the population, killing between one-quarter and one-third of all people [24]. This catastrophic event marked one of the deadliest outbreaks in human history.

The expression refers to the symbolic meaning of the adjective (e.g., lugubrious and appalling). Accordingly, the Black Death does not date from the 14th century and is entirely anachronistic [21]. During the Middle Ages, the plague was often referred to as the “great mortality”, “bumps disease”, and “epidemic”. Moreover, several regions experienced multiple disease outbreaks, as a chronicler from Orvieto [25] noted: “The first general plague occurred in 1348 and was the strongest”. Then, this author adds the following: “Second plague, 1363; Third plague, 1374; Fourth plague, 1383; Fifth plague, 1389; Sixth Plague, 1410”. Biraben perfectly summarized the endemicity of the disease as follows: “From that point onward, and until 1670, the plague raged every year in Europe, sometimes in vast territories, sometimes only in certain localities, but without skipping a single annual link in this long and painful chain” [26].

From 1347 until the mid-17th century, the plague persisted in Western European societies. After 1670, its occurrence became less frequent, though notable outbreaks continued into the 18th century. These included the episodes in Marseille and Provence from 1720 to 1722 [27].

The plague had significant social repercussions. It ended the Hundred Years’ War between the French and English kingdoms, which disrupted economic expansion across the continent and impacted the workforce [28]. Pinpointing the exact end of the second pandemic is challenging. Antoine-Jean Gros, for instance, depicted plague-stricken soldiers of Bonaparte during the Egyptian Campaign in Jaffa in 1799 [29]. The plague remained present in Malta in 1813 [17], in Tunisia between 1818 and 1820 [30], and in Egypt between 1834 and 1835 [31,32]. The 1839 epidemic in Constantinople [33] is considered the final occurrence of the second pandemic in Europe [23] (Signoli, 2012).

Although several centuries separate the first and second pandemics, the chronological gap between the second and third pandemics is less distinct. The third pandemic’s first manifestation occurred in 1772 in Yunnan Province, southwestern China [33]. Europe continued to experience outbreaks attributed to the second pandemic.

#### 3.1.3. Third Pandemic (1772 to 1945)

The third pandemic plague likely originated in the Yunnan province of southwestern China, with records indicating its presence as early as 1772 in Dali [34,35]. The plague became endemic in the region during the 1850s and 1860s [36], thriving amidst the movements of troops and refugees during the Mohammedan rebellion. By March 1894, the plague had reached Canton, caused over 60,000 deaths, and rapidly spread to Hong Kong by May of the same year [37]. Between 1899 and 1900, the plague spread across all continents, affecting Asia (India in 1896 and Japan in 1899), the Middle East (Saudi Arabia and Turkey in 1897) [23,32], Africa (Madagascar in 1898), Oceania (Brisbane in 1899 and Sydney in 1900), Europe (Lisbon in 1899 and Glasgow in 1900), and North and South America (Brazil, Paraguay, and Honolulu in 1899, and San Francisco in 1900) [23,34,36]. This pandemic was recorded in over 100 countries [34].

The rapid spread of this disease along steamboat and train routes alarmed European health authorities, prompting an emergency meeting in Venice in 1897 to control the potential return of the plague to Europe [23]. Despite enhanced quarantine measures, the bubonic plague returned to Lisbon in 1899 and Glasgow in 1900, resulting in 37 deaths [29,38].

The plague also reached regions previously unaffected, including South America, the United States, South Africa, and Australia. *Y. pestis* established permanent foci in all these regions except Australia. Comparing the outcomes of the third pandemic plague in the United States and Australia reveals that the disease entered both countries around the same period (San Francisco in 1896 and 1900 [39], and Brisbane and Sydney in 1899 and 1900) via maritime routes. In the U.S., the plague persisted in harbor towns until the 1930s. It became an inland endemic infection by the 1950s, affecting rodent populations and establishing endemicity in the semi-arid southern states [40].

The plague entered Brisbane Harbor in Australia in 1899 and Sydney a year later. Affected districts were isolated, the inhabitants were evacuated, and over 100,000 rats were killed to prevent the disease’s spread [39]. Unlike in the U.S., the reasons why the plague did not become permanently established in Australia remain unclear and may be linked to the inability of *Y. pestis* to establish permanent foci in Australian soil [12,38,41].

From 1910 to 1911, a significant presumed pneumonic plague outbreak in Manchuria caused approximately 60,000 deaths [42]. The Mukden Conference established a system of international cooperation to combat the plague during the rise of steamboat and railroad transportation [43]. In France, the final cases were recorded in Marseille from 1919 to 1920 [27] and in Paris in June 1920 during an episode known as “Ragman’s plague” [44]. This outbreak, which co-occurred with the Spanish influenza, confused the two diseases. Genomic analyses indicated that single introductions followed by regional dispersion were typical in Brazil. Following World War II, the last plague in Europe occurred in 1945 in Corsica and Italy and in 1947 in Kaliningrad [23,38]. The third pandemic resulted in more than 26 million cases and over 12 million deaths in India and China alone [36].

### 3.2. Most Recent Emerging Epidemics Caused by Antimicrobial-Resistant Bacteria

An emerging infection appears for the first time in a population or spreads to a new geographic area. Zoonotic transmission, where pathogens move from animals to humans, has been a significant mechanism through which emerging infections throughout history have afflicted humans. The species of animals that harbor the pathogens, the nature of human interaction with these animals, and the frequency of these interactions are critical factors that influence the risk of zoonotic transmission [45].

The spread of several infectious diseases, such as tuberculosis, malaria, and cholera, to broader geographic areas is raising significant health concerns. These diseases are spreading more widely due to drug resistance, mosquito vectors’ insecticide tolerance, poor sanitation, land use and climate change, and increased human mobility and travel. Cholera outbreaks have also been reported in regions affected by natural disasters such as earthquakes and floods [45].

Research has identified particularly dangerous clones of specific pathogens. For instance, *Clostridium difficile*’s O27 clone, which emerged a few years ago, is highly infectious and deadly. Detecting this clone is essential for implementing adequate measures. Similarly, the Beijing genotype of *Mycobacterium tuberculosis*, identified as an epidemic clone capable of carrying multiple resistance determinants, now represents 10% of *M. tuberculosis* isolates globally [46].

In the case of *Neisseria meningitidis*, the genomic analysis revealed a surprising finding: the clone currently circulating in the UK can be either serotype A or C, which challenges our classification and may affect vaccine strategies. Cholera genomics studies have shown that among the vast population of *Vibrio cholerae* in the environment, only a few clones have the potential to spread and cause epidemics and pandemics. The cholera epidemic in Haiti, for instance, was caused by a virulent clone already detected in Nepal and carried by Nepalese UN soldiers, highlighting the importance of identifying specific clones in outbreak management [47].

Infectious diseases continue to pose significant threats, as pathogens can spread rapidly through global trade and travel. Therefore, global surveillance programs are essential to detect and identify pathogen spillovers from animals to humans and control water-borne and vector-borne diseases. Non-pharmaceutical and pharmaceutical measures are needed to prevent and control these infections and limit their spread in human populations [46].

Genomics has significantly enhanced our understanding of bacterial epidemics, revealing that most epidemics are caused by individual clones that require specific identification and target preventive measures. As a result, surveillance programs are crucial to rapidly detect emerging pathogens with zoonotic transmission potential at the animal–human interface and control their spread from endemic to non-endemic regions [46,48].

### 3.3. Bacterial Resistance

Bacterial resistance occurs when bacteria undergo evolutionary changes that make them less susceptible to antibiotics, making infections more difficult to treat and cure. Antibiotic mechanisms of action are classified into four groups: (1) inhibition of cell wall turnover, (2) inhibition of protein synthesis, (3) inhibition of DNA replication, and (4) antimetabolic (Figure 3). The problem of antibiotics is that they target proteins. In this context, antibacterial-resistant genes (ARGs) can arise or be selected in a bacterial population under drug, environmental, or host pressure [49]. Suppose the selection of resistance in a particular population was not enough. In that case, bacteria can transfer ARGs horizontally among organisms via MGEs, enhancing this problem and turning ARG dissemination into one of the most serious public health issues today [17,18,50,51].

Bacterial resistance is a growing problem worldwide, and antibiotics are becoming less and less effective in treating infectious diseases. The World Health Organization (WHO) fears bacterial resistance will reach new heights due to the shortage of effective antibiotics [52]. The increasingly intense appearance of multi-resistant bacteria to antibiotics has become a significant challenge for treating infections, resulting in the global rise of infectious diseases, especially nosocomial infections [53].

Numerous mechanisms of bacterial resistance to antibiotics arise and spread globally, potentially interfering with the therapeutic success of infectious diseases [19]. Environmental stressors can increase the emergence of bacterial mutations that confer antibiotic resistance. It has been shown that in addition to the antibiotics themselves, other chemicals such as heavy metals, biocides, disinfection products, and non-chemical environmental stressors such as ultraviolet radiation contribute to this increase. Resistance mutations are more likely to occur when bacteria are subjected to sublethal or subinhibitory levels of these stressors, raising a global environmental concern [54,55].

The great proof that the evolution of resistance occurs independently of anthropic action and has been occurring for millions of years is the findings of resistant genes in bacteria from pristine locations, including permafrost and glacial soil [56,57].

Some studies have shown that permafrost can be a reservoir for ARGs. A metagenomics study evaluated the presence of ARGs and pathogenic antibiotic-resistant bacteria (PARB) in Arctic permafrost. The results of this study demonstrated that, in general, 70 ARGs were found against 18 classes of antimicrobials and 15 PARB. The ARGs found in the Arctic differ from the genes in the reference databases used for comparison. These findings suggest that Arctic permafrost can be considered a potential reservoir for ARGs and PARB.

Using culture-independent/metagenomic analysis or bacterial isolation from permafrost areas, other reports demonstrated that there are resistant mechanisms/genes against diverse types of antibiotics targeting different processes in the bacterium, such as cell wall synthesis (ampicillin, penicillin, carbenicillin, vancomycin, teicoplanin) protein synthesis (tetracycline, doxycycline, sisomicin, gentamicin, streptomycin, chloramphenicol, amikacin) DNA replication (nalidixic acid, norfloxacin, novobiocin), and the folate pathway (trimethoprim, sulfonamide) [17,18,56,58,59,60,61,62].

These data prove that antibiotic resistance precedes the appearance and large-scale production of antimicrobial drugs, which began less than 100 years ago. Most antibiotics are natural weapons used in inter-microbial interactions; therefore, it is not surprising that antibiotic resistance is a natural process. However, bacteria from permafrost can harbor resistant genes to semi-synthetic antibiotics, too, such as amikacin, a “modern drug” [62].

The environment and climate change influence the emergence of antibiotic-resistant bacteria once the permafrost has melted, creating a source for ARGs. Bacteria unearthed from melting could interact and transfer antibiotic resistance genes to another non-psychrophilic/non-glacial but pathogenic bacterium, turning these pathogens that are susceptible to some antibiotics into resistant pathogens. In addition, ARGs could be transferred to other microbes from newly unearthed bacteria, and naked free DNA from the environment can also be transferred, which may also emerge from the thawing of permafrost [49]. In this context, these genes can potentially threaten humanity if they are released by the melting of permafrost due to global warming and spread to other regions of the planet [63].

Permafrost thawing due to climate change poses a significant environmental threat, potentially leading to the resurgence of ancient pathogens and the spread of resistant genes [57]. As permafrost thaws, long-dormant viruses and bacteria preserved for millennia are exposed and could potentially infect humans and animals [58,59,60,61,62]. This phenomenon has already been observed with the revival of viruses from permafrost dating back thousands of years [60]. Environmental changes, such as rising temperatures and increased flooding, exacerbate this problem by creating favorable conditions for the spread of these pathogens [61]. Additionally, climate change can contribute to the proliferation of antibiotic-resistant bacteria by altering ecosystems and increasing selective pressures. Urgent scientific efforts are needed to study the effects of climate change on microbial resistance and develop strategies to mitigate these emerging threats [62]. Understanding these dynamics is crucial for safeguarding public health and ensuring global stability in the face of environmental change.

The urgency of combatting antimicrobial resistance (AMR) cannot be overstated. Implementing preventive measures, such as improved hygiene and infection control, is paramount in stopping the spread of resistant bacteria [58,60]. Enhancing antibiotic stewardship is crucial as well as ensuring that these vital medications are used wisely and only when necessary. Investing in new antimicrobial drug development is essential to stay ahead of evolving pathogens. However, addressing AMR requires more than individual efforts; it demands global cooperation. By working together and sharing knowledge, resources, and strategies, it is possible to mitigate the devastating impact of AMR on public health worldwide. It is a collective responsibility to ensure that future generations have effective treatments against infections.

It is imperative that policymakers, researchers, and healthcare professionals come together to prioritize the fight against antimicrobial resistance (AMR). This requires a significant increase in funding dedicated to research on new antibiotics, alternative therapies, and resistance mechanisms. Additionally, developing and implementing comprehensive policies are crucial to controlling the spread of resistant bacteria. These policies should include stricter regulations on antibiotic use in human medicine and agriculture, improved surveillance systems to monitor resistance trends, and enhanced infection prevention and control measures. Collaboration across sectors and countries is essential to effectively address this global health threat. By taking decisive action now, we can ensure the continued efficacy of antibiotics, protect public health, and safeguard future generations from the devastating consequences of AMR.

### 3.4. Antimicrobial Peptides and Recent Applications

In recent years, antimicrobial peptides (AMPs) have garnered significant interest from scientists, health professionals, and pharmaceutical companies due to their therapeutic potential. These are low molecular weight proteins with a broad spectrum of antimicrobial and immunomodulatory activities against infectious bacteria (both Gram positive and Gram negative), viruses, and fungi [64].

AMPs offer several advantages over conventional antibiotics, including a slower emergence of resistance, broad-spectrum antibiofilm activity, and the ability to modulate the host immune response [65]. Their pharmacodynamics and the fact that multicellular hosts naturally use them in synergistic combinations limit the likelihood of bacterial resistance developing in nature [66]. From the peptides listed in the Data Repository of Antimicrobial Peptides (DRAMP, http://dramp.cpu-bioinfor.org/ accessed on 10 August 2024), 70 have progressed into the drug development pipeline, with 27 in clinical trials and 34 at the preclinical stage. The FDA has approved eight AMPs to date [55]. Further research is essential to better understand their natural biology and evolution, minimizing collateral harm and addressing the current antibiotic resistance crisis.

Looking to the future, innovative approaches to drug development hold immense promise, particularly with AMPs. These natural or synthetic molecules exhibit a broad spectrum of antimicrobial activity. They can target a wide range of pathogens, including drug-resistant bacteria. AMPs can be the foundation for developing new antibiotics that bypass traditional resistance mechanisms. Additionally, advancements in synthetic biology could enable the design of custom AMPs tailored to specific pathogens, thereby enhancing their efficacy. Furthermore, exploring nanotechnology for targeted drug delivery and employing AI-driven drug discovery processes can streamline the identification of novel therapeutic compounds. Together, these cutting-edge strategies could revolutionize the fight against infectious diseases and significantly mitigate the threat of antimicrobial resistance in the future [60,61,62,63,64].

## 4. Conclusions

This review highlights the history of bacteria that have caused pandemics and led to millions of deaths and emphasizes the critical importance of addressing bacterial resistance to drugs, particularly the emergence of new pathogens in the context of permafrost thawing due to climate change. As these pathogens are reintroduced into the environment, there is an increased risk of infectious diseases that humans and animals have not encountered for centuries, leading to potential outbreaks and global health crises.

Moreover, this review highlights the urgency of searching for new drugs to combat these emerging threats. Research involving peptides, specifically AMPs, is fundamental in this quest. AMPs offer a promising alternative for developing new antibiotics and antiviral agents due to their broad-spectrum activity and ability to target resistant bacteria. The exploration of AMPs and other innovative approaches is crucial in staying ahead of evolving microbial threats and ensuring the continued efficacy of medical treatments. Overall, this review underscores the necessity of proactive scientific research and policy measures to address the complex challenges of environmental changes and the resurgence of ancient pathogens.

## Figures and Tables

**Figure 1 microorganisms-13-00457-f001:**
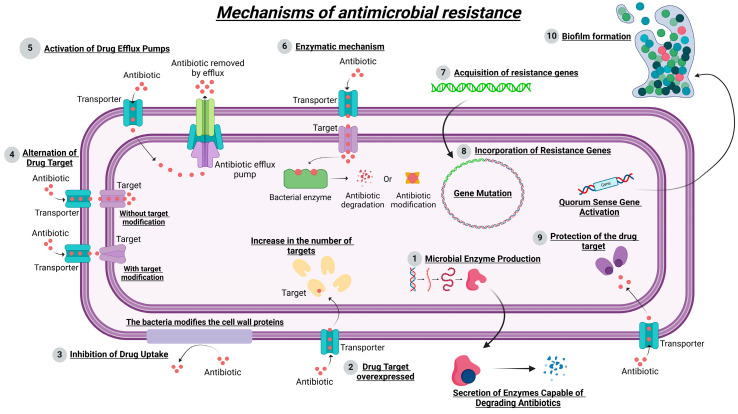
Mechanisms involved in bacterial resistance to antibiotics.

**Figure 2 microorganisms-13-00457-f002:**
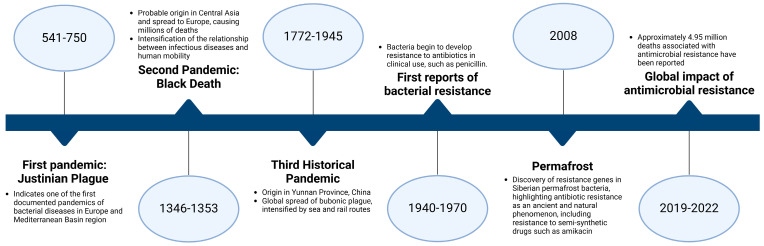
Timeline of events involving bacteria throughout human history.

**Figure 3 microorganisms-13-00457-f003:**
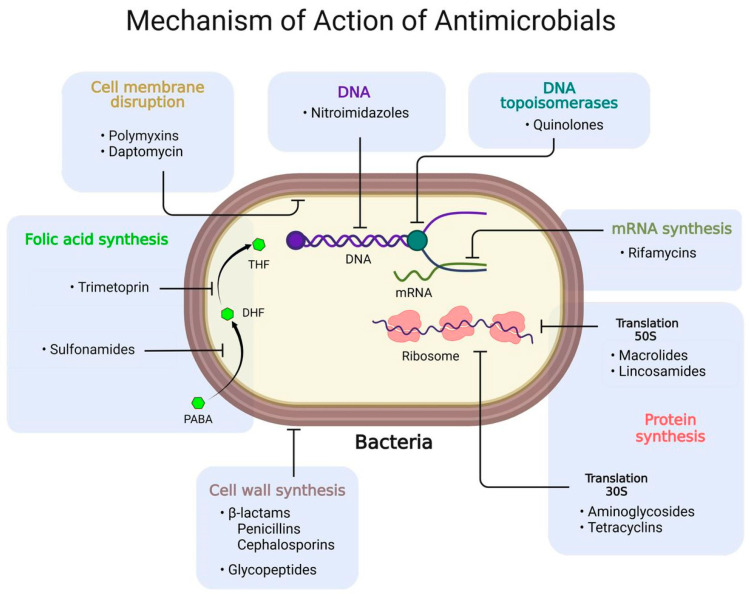
Mechanisms of action of antibiotics.

## Data Availability

No new data were created or analyzed in this study.

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
