# Peer review of "Pandemic Events Caused by Bacteria Throughout Human History and the Risks of Antimicrobial Resistance Today"

_microorganisms, 2025, doi:10.3390/microorganisms13020457_

Round 1
Reviewer 1 Report
Comments and Suggestions for Authors
The manuscript entitled „ Antimicrobial resistance in bacteria throughout human history and today” is a review manuscript, that gives an overview about antibiotic resistance. The topic itself is an interesting one, however, the manuscript should be strengthened.
Comments
1) Abstract is very general. Specific data should be added to the abstract.
2) The main text of this manuscript is unequal. Very general data of antibiotic resistance (e.g. resistance mechansisms) are described. However, a very detailed description of plague is given. But other medically important bacteria are briefly described (e.g. Neisseria meningitidis, Vibrio cholerae, Clostridium difficile (current name: Clostridioides difficile), Mycobacterium tuberculosis.
3) I suggest to authors to describe other medically important bacteria including typical antibiotic resistantance mechanisms of each bacterium.
4) I also suggest to add some diagrams to demonstrate development of antibiotic resistant bacteria in chronological way. In the current manuscript only plague is demontrated in timeline.
Author Response
Reviewers' comments:
Reviewer #1 - Comments to the Author
The manuscript entitled „ Antimicrobial resistance in bacteria throughout human history and today” is a review manuscript, that gives an overview about antibiotic resistance. The topic itself is an interesting one, however, the manuscript should be strengthened.
Authors’ General Response
Dear Reviewer #1
Thank you for agreeing to review our manuscript. We are glad about your thoughts about our work. We are thankful for your comments on our manuscript. Indeed, your suggestion did help to bring this manuscript to a higher level. You can be sure that we did our best in this manuscript to address all the compliments that you made.
Point by Point Response to Reviewer #1
Reviewer #1 - Comment 1
- Abstract is very general. Specific data should be added to the abstract.
Authors’ Response 1
Following you request, we have improved the abstract.
Reviewer #1 - Comment 2
- The main text of this manuscript is unequal. Very general data of antibiotic resistance (e.g. resistance mechansisms) are described. However, a very detailed description of plague is given. But other medically important bacteria are briefly described (e.g. Neisseria meningitidis, Vibrio cholerae, Clostridium difficile (current name: Clostridioides difficile), Mycobacterium tuberculosis.
Authors’ Response 2
Dear reviewer #1,
We understand your concern. However, the idea of our manuscript is to discuss the pandemic caused by bacteria and the risks of new pandemic events. Based on your comment, we realize that the title of our study did not reflect what we try to discuss. Therefore, we change the title to: “Pandemic events caused by bacteria throughout human history and the risks of antimicrobial resistance today”
Reviewer #1 - Comment 3
- I suggest to authors to describe other medically important bacteria including typical antibiotic resistantance mechanisms of each bacterium.
Authors’ Response 3
Dear reviewer #1
This is not the topic of our study. If we added all other bacteria that are medically important with all mechanisms of resistance, the study will never have end.
Reviewer #1 - Comment 4
- I also suggest to add some diagrams to demonstrate development of antibiotic resistant bacteria in chronological way. In the current manuscript only plague is demontrated in timeline.
Authors’ Response 4
We focused on plague because it is the most important bacteria discussed in the pandemic events that occurred in human history.
Reviewer 2 Report
Comments and Suggestions for Authors
1. The manuscript mentions antimicrobial resistance (AMR) as a major issue but doesn't explain the scope or urgency of the problem clearly. It would be helpful to quantify the global impact of AMR, such as the number of deaths or the financial burden it causes each year.
2. The manuscript could be improved by explicitly stating the research question or hypothesis. For example: "This review aims to evaluate the impact of antimicrobial resistance on the emergence of reemerging infections."
3. The methodology is briefly mentioned, but it should be expanded to include how the literature search was conducted. For example, you could specify the time period covered, the number of studies reviewed, and the inclusion criteria.
4. The results statement is broad. Try to provide more specific findings or trends from the review, such as the most common resistant pathogens or the mechanisms through which resistance occurs.
5. Briefly mention the novelty of this review. What differentiates this study from other reviews on AMR? Is there a new perspective or a focus on certain bacterial strains or mechanisms?
6. The manuscript ends with a statement on the importance of peptide research but does not offer a clear conclusion or call to action. Consider emphasizing the potential next steps, such as the need for more research or the development of specific therapies.
7. The manuscriptfeels slightly disjointed, moving from AMR to reemerging infections and then to peptide research. Try to create a more cohesive narrative by connecting these points with smoother transitions, demonstrating how they relate to each other.
8. The opening statement about the 700 thousand infections could benefit from more context. For example, you could specify that these cases are a result of antimicrobial-resistant bacteria (AMRB), emphasizing the growing global concern surrounding AMR.
9. The introduction mentions that AMRB infections in 2020-2022 are "superior to other diseases." This needs more clarity. For instance, is it referring to the number of cases or the mortality rate? Consider rephrasing to make the comparison more specific, such as: "AMRB infections caused more cases than other leading diseases, including viral infections and chronic conditions like cancer and diabetes."
10. The statement about the economic impact (2-3.5% of GDP) could be expanded to provide more details, such as the estimated global cost of AMR and how it impacts healthcare systems and society as a whole.
11. The sentence “Although most bacterial are potential harmless, infection caused by AMRB are hard to treat requiring a multiple pharmacological approach…” is a bit awkward. It can be reworded for better readability, e.g., "Although many bacteria are harmless under normal circumstances, infections caused by AMRB are difficult to treat and often require a multi-drug approach, leading to severe disease and death."
12. The sentence "Based on that, it is urgently necessary the development of efficient methods to detect AMRB infections" is somewhat vague. Consider clarifying the focus of the study—whether it involves developing new detection methods, understanding the mechanisms of resistance, or exploring treatment strategies.
13. The connection between historical developments in bacterial resistance and current issues could be strengthened. For instance, it would be beneficial to briefly mention how the misuse of antibiotics (historically and today) has accelerated resistance and its consequences, making the reader understand the long-term problem.
14. The search strategy should include more detail on how keywords were used in the databases. For example, were Boolean operators or specific search filters applied to refine the results? Adding this will provide transparency to the review process.
15. The inclusion criteria are well defined, but the exclusion criteria are not mentioned. It would be helpful to specify what types of studies were excluded (e.g., non-peer-reviewed articles, studies not related to human health, or those outside the 5-year publication window).
16. It would be helpful to indicate how many articles were initially identified and how many were ultimately selected for inclusion in the review. This gives readers a sense of the scope and rigor of the literature review process.
17. The statement "The publications in the last 5 years were individually searched and selected by investigators" should be expanded to explain whether the investigators reviewed the studies independently and how disagreements were resolved (e.g., through consensus or third-party review).
18. Consider briefly explaining why MEDLINE and PUBMED were chosen as the primary databases. Were other databases considered or excluded for any specific reason? Justifying the choice can strengthen the method's credibility.
19. Expand on the broader implications of antimicrobial resistance (AMR) for public health beyond permafrost thawing. Discuss how AMR threatens the effectiveness of existing antibiotics and the global health crisis it could lead to if left unaddressed.
20. Highlight the urgency of combating AMR by implementing preventive measures, improving antibiotic stewardship, and investing in new antimicrobial drug development. Mention the importance of global cooperation in addressing AMR to reduce its impact on public health.
21. Offer specific suggestions for the future, such as the potential role of antimicrobial peptides (AMPs) and other innovative approaches to drug development. This could include promoting further research, clinical trials, and the commercialization of novel antimicrobial agents.
22. Expand on the environmental aspect, particularly the thawing of permafrost, as it could lead to the resurgence of ancient pathogens and resistant genes. Discuss how environmental changes exacerbate the problem and necessitate urgent scientific efforts to study the effects of climate change on microbial resistance.
23. Include a call to action for policymakers, researchers, and healthcare professionals to prioritize the fight against AMR, both in terms of funding research and creating new policies aimed at controlling the spread of resistant bacteria.
24. By expanding the conclusion to include these points, it will provide a more comprehensive summary of the review's findings, while emphasizing the importance of continued research, global collaboration, and proactive measures in combating AMR.
Author Response
Reviewer #2
Response to Reviewer #2
Reviewer #2 - Comment 1
- The manuscript mentions antimicrobial resistance (AMR) as a major issue but doesn't explain the scope or urgency of the problem clearly. It would be helpful to quantify the global impact of AMR, such as the number of deaths or the financial burden it causes each year.
Authors’ Response 1
The information you required was added into the end of introduction.
Reviewer #2 - Comment 2
- The manuscript could be improved by explicitly stating the research question or hypothesis. For example: "This review aims to evaluate the impact of antimicrobial resistance on the emergence of reemerging infections."
Authors’ Response 2
The information was added to the end of the manuscript. Thank you for that comment.
Reviewer #2 - Comment 3
- The methodology is briefly mentioned, but it should be expanded to include how the literature search was conducted. For example, you could specify the time period covered, the number of studies reviewed, and the inclusion criteria.
Authors’ Response 3
Dear reviewer,
All this information is already in the methodology.
Reviewer #2 - Comment 4
- The results statement is broad. Try to provide more specific findings or trends from the review, such as the most common resistant pathogens or the mechanisms through which resistance occurs.
Authors’ Response 4
We understand your concern. However, the idea of our manuscript is to discuss the pandemic caused by bacteria and the risks of new pandemic events. Based on your comment, we realize that the title of our study did not reflect what we try to discuss. Therefore, we change the title to: “Pandemic events caused by bacteria throughout human history and the risks of antimicrobial resistance today”
Reviewer #2 - Comment 5
- Briefly mention the novelty of this review. What differentiates this study from other reviews on AMR? Is there a new perspective or a focus on certain bacterial strains or mechanisms?
Authors’ Response 5
Thank you for the comment. As you requested, the information was added in the last paragraph of the introduction.
Reviewer #2 - Comment 6
- The manuscript ends with a statement on the importance of peptide research but does not offer a clear conclusion or call to action. Consider emphasizing the potential next steps, such as the need for more research or the development of specific therapies.
Authors’ Response 6
Dear reviewer,
The last statement is just to show that peptides could be a good therapeutic alternative. Our research group has other manuscripts discussing this point you addressed.
Reviewer #2 - Comment 7
- The manuscriptfeels slightly disjointed, moving from AMR to reemerging infections and then to peptide research. Try to create a more cohesive narrative by connecting these points with smoother transitions, demonstrating how they relate to each other.
Authors’ Response 7
We respectfully disagree from you in this topic. The manuscript follows one flow about the pandemicàemergent bacteriaàbacterial resistanceàpeptides as solution to resistance.
Reviewer #2 - Comment 8
- The opening statement about the 700 thousand infections could benefit from more context. For example, you could specify that these cases are a result of antimicrobial-resistant bacteria (AMRB), emphasizing the growing global concern surrounding AMR.
Authors’ Response 8
Thank you for this comment. It was fixed. Please see the new text added.
Reviewer #2 - Comment 9
- The introduction mentions that AMRB infections in 2020-2022 are "superior to other diseases." This needs more clarity. For instance, is it referring to the number of cases or the mortality rate? Consider rephrasing to make the comparison more specific, such as: "AMRB infections caused more cases than other leading diseases, including viral infections and chronic conditions like cancer and diabetes."
Authors’ Response 9
Thank you for this comment. It was fixed.
Reviewer #2 - Comment 10
- The statement about the economic impact (2-3.5% of GDP) could be expanded to provide more details, such as the estimated global cost of AMR and how it impacts healthcare systems and society as a whole.
Authors’ Response 10
Thank you for this comment. It was fixed in text.
Reviewer #2 - Comment 11
- The sentence “Although most bacterial are potential harmless, infection caused by AMRB are hard to treat requiring a multiple pharmacological approach…” is a bit awkward. It can be reworded for better readability, e.g., "Although many bacteria are harmless under normal circumstances, infections caused by AMRB are difficult to treat and often require a multi-drug approach, leading to severe disease and death."
Authors’ Response 11
Thank you for this comment and suggestion. We follow strictly your suggestion.
Reviewer #2 - Comment 12
- The sentence "Based on that, it is urgently necessary the development of efficient methods to detect AMRB infections" is somewhat vague. Consider clarifying the focus of the study—whether it involves developing new detection methods, understanding the mechanisms of resistance, or exploring treatment strategies.
Authors’ Response 12
Thank you for this comment and suggestion. We rewrite the sentence.
Reviewer #2 - Comment 13
- The sentence "Based on that, it is urgently necessary the development of efficient methods to detect AMRB infections" is somewhat vague. Consider clarifying the focus of the study—whether it involves developing new detection methods, understanding the mechanisms of resistance, or exploring treatment strategies.
Authors’ Response 13
We agree with you. A new paragraph was produced.
Reviewer #2 - Comment 14
- The search strategy should include more detail on how keywords were used in the databases. For example, were Boolean operators or specific search filters applied to refine the results? Adding this will provide transparency to the review process.
Authors’ Response 14
New information was added to the methodology.
Reviewer #2 - Comment 15
- The inclusion criteria are well defined, but the exclusion criteria are not mentioned. It would be helpful to specify what types of studies were excluded (e.g., non-peer-reviewed articles, studies not related to human health, or those outside the 5-year publication window).
Authors’ Response 15
New information was added to the methodology.
Reviewer #2 - Comment 15
- The inclusion criteria are well defined, but the exclusion criteria are not mentioned. It would be helpful to specify what types of studies were excluded (e.g., non-peer-reviewed articles, studies not related to human health, or those outside the 5-year publication window).
Authors’ Response 15
New information was added to the methodology.
Reviewer #2 - Comment 16
- It would be helpful to indicate how many articles were initially identified and how many were ultimately selected for inclusion in the review. This gives readers a sense of the scope and rigor of the literature review process.
Authors’ Response 16
New information was added to the methodology.
Reviewer #2 - Comment 17
- The statement "The publications in the last 5 years were individually searched and selected by investigators" should be expanded to explain whether the investigators reviewed the studies independently and how disagreements were resolved (e.g., through consensus or third-party review).
Authors’ Response 17
Dear reviewer,
The studies older than five years were automatically excluded from study.
Reviewer #2 - Comment 18
- Consider briefly explaining why MEDLINE and PUBMED were chosen as the primary databases. Were other databases considered or excluded for any specific reason? Justifying the choice can strengthen the method's credibility.
Authors’ Response 18
Because they are huge databases and repository of studies. However, as requested by you, the information was added to the study.
Reviewer #2 - Comment 19
- Expand on the broader implications of antimicrobial resistance (AMR) for public health beyond permafrost thawing. Discuss how AMR threatens the effectiveness of existing antibiotics and the global health crisis it could lead to if left unaddressed.
Authors’ Response 19
Dear reviewer,
This is not the focus of the manuscript.
Reviewer #2 - Comment 20
- Highlight the urgency of combating AMR by implementing preventive measures, improving antibiotic stewardship, and investing in new antimicrobial drug development. Mention the importance of global cooperation in addressing AMR to reduce its impact on public health.
Authors’ Response 20
Dear reviewer,
A new paragraph was added at the end of the topic of bacterial resistance. Thank you.
Reviewer #2 - Comment 21
- Offer specific suggestions for the future, such as the potential role of antimicrobial peptides (AMPs) and other innovative approaches to drug development. This could include promoting further research, clinical trials, and the commercialization of novel antimicrobial agents.
Authors’ Response 21
Dear reviewer,
A new paragraph was added at the end of the topic of antimicrobial peptides and application. Thank you.
Reviewer #2 - Comment 22
- Expand on the environmental aspect, particularly the thawing of permafrost, as it could lead to the resurgence of ancient pathogens and resistant genes. Discuss how environmental changes exacerbate the problem and necessitate urgent scientific efforts to study the effects of climate change on microbial resistance.
Authors’ Response 22
Dear reviewer,
The new paragraph was added in the topic.
Reviewer #2 - Comment 23
- Include a call to action for policymakers, researchers, and healthcare professionals to prioritize the fight against AMR, both in terms of funding research and creating new policies aimed at controlling the spread of resistant bacteria.
Authors’ Response 23
Dear reviewer,
The new paragraph was added to the topic.
Reviewer #2 - Comment 24
- By expanding the conclusion to include these points, it will provide a more comprehensive summary of the review's findings, while emphasizing the importance of continued research, global collaboration, and proactive measures in combating AMR.
Authors’ Response 24
Dear reviewer,
The new paragraph was added to the topic.
Reviewer 3 Report
Comments and Suggestions for Authors
General comments
The manuscript needs extensive English language editing. I suggest the authors use a native English speaker for this purpose.
Introduction
"There have been 700 thousand cases of infection caused by antimicrobial resistance bacteria (AMRB) in 2020-2022." Where? provide a reference.
"Infections caused by AMRB an investment of 2 to 3.5% of the gross domestic product" This doesn't make sense. Rephrase
"It is estimated that more than 10 million deaths will occur annually" Where? When?
"Although most bacterial are potential harmless, infection caused by AMRB are hard to treat requiring a multiple pharmacological approach with different antibiotics at the same time and usually led to a severe disease and death" This does not make sense. Rephrase.
"According to a systematic review in 2019, there was an estimation of 4,95 million new deaths caused by AMRB in 2019." Provide a reference
The last paragraph of the introduction should be rephrased entirely.
Materials and methods
The search terms are not sufficient to provide sufficient articles for such a review. Also, what criteria determined review rigour of journal? How many articles met the inclusion criteria?
Results and discussion
Figure 1 does not show anything related to the results. Also, figure two is similar to what many other authors have reported previously. Here, the authors should instead plot a trend curve that shows that the number of infections increased after the emergence of AMR.
Sections 3.1.1 to 3.1.3 do not say anything about AMR. The information reported here has been extensively reviewed in many papers on the history of pandemics. If the authors cannot provide a link between these plagues and AMR, then the sections provide no new information. The same applies to Section 3.2.
Section 3.3. Bacterial resistance is not always due to evolutionary changes. Also, bacteria can acquire resistance through congugation, transformation or transduction.
Section 3.4: AMPs are only one option to fight AMR today. There is also Phage theraphy. The authors should also bring in the one health concept and how addressing AMR in all three sectors is important.
Specific comments
Page 1, Line 3: Delete "data"
Page 1, Line 4-6: "Results showed that antimicrobial-resistant genes and antibiotic-resistant bacteria in organisms cause critical infectious diseases." This does not make sense. rephrase. Also, change "antimicrobial-resistant genes" to " antimicrobial resistance genes"
Comments on the Quality of English LanguageThe manuscript needs extensive language editing
Author Response
Response to Reviewer #3
Reviewer #3 - Comment 1
The manuscript needs extensive English language editing. I suggest the authors use a native English speaker for this purpose.
Authors’ Response 1
As requested by you, the manuscript was revised native English speaker.
Reviewer #3 - Comment 2
Introduction
"There have been 700 thousand cases of infection caused by antimicrobial resistance bacteria (AMRB) in 2020-2022." Where? provide a reference.
"Infections caused by AMRB an investment of 2 to 3.5% of the gross domestic product" This doesn't make sense. Rephrase
"It is estimated that more than 10 million deaths will occur annually" Where? When?
"Although most bacterial are potential harmless, infection caused by AMRB are hard to treat requiring a multiple pharmacological approach with different antibiotics at the same time and usually led to a severe disease and death" This does not make sense. Rephrase.
"According to a systematic review in 2019, there was an estimation of 4,95 million new deaths caused by AMRB in 2019." Provide a reference
The last paragraph of the introduction should be rephrased entirely.
Authors’ Response 2
The introduction was rewritten to improve it.
Reviewer #3 - Comment 3
Materials and methods
The search terms are not sufficient to provide sufficient articles for such a review. Also, what criteria determined review rigour of journal? How many articles met the inclusion criteria?
Authors’ Response 3
All this information is now in the revised version of the methodology.
Reviewer #3 - Comment 4
Figure 1 does not show anything related to the results. Also, figure two is similar to what many other authors have reported previously. Here, the authors should instead plot a trend curve that shows that the number of infections increased after the emergence of AMR.
Authors’ Response 4
Dear reviewer #3,
The figures is related to the topic of bacterial resistance. Many other figures of others studies are similar. Today, with the volume of paper published is pretty much impossible to perform figures for reviews that nobody has done similar before.
Reviewer #3 - Comment 5
Sections 3.1.1 to 3.1.3 do not say anything about AMR. The information reported here has been extensively reviewed in many papers on the history of pandemics. If the authors cannot provide a link between these plagues and AMR, then the sections provide no new information. The same applies to Section 3.2.
Authors’ Response 5
Dear reviewer, we disagree with you. We try to perform a history, to let readers understand the past about how bacteria threatened humanity and now see the problem of resistance. Topic 3.2 is to warn, that if nothing is done to change this situation dark times could come back.
Again,
Reviewer #3 - Comment 6
Methods: A clear explanation of frameworks, inclusion, and exclusion criteria should be given with search terms. Who did quality control check, how many scientists were involved in it. So far, methods section seems very vague and generic.
Authors’ Response 6
Dear reviewer, all the criteria were discussed in the methods. All authors have contributed to that part. This was the most important part of the work. We have done our best to reach it.
Most of the information in review about bacterial resistance are similar. However, each work is unique and relevant, because e a new point of view, and report experiences of research that are doing.
Reviewer #3 - Comment 7
Section 3.3. Bacterial resistance is not always due to evolutionary changes. Also, bacteria can acquire resistance through congugation, transformation or transduction.
Authors’ Response 7
Dear reviewer #3,
All these processes lead to the evolution of cells involved in it. Any cell thar going through this process becomes a new individual with a new feature (e.g., resistance to drugs) allowing survival in adverse environment.
Reviewer #3 - Comment 8
Section 3.4: AMPs are only one option to fight AMR today. There is also Phage theraphy. The authors should also bring in the one health concept and how addressing AMR in all three sectors is important.
Authors’ Response 8
Dear reviewer,
We understand that. However, we decided to talk about AMPs. To talk about each other options to fight AMR, we be necessary another study.
Reviewer #3 - Comment 9
Specific comments
Page 1, Line 3: Delete "data"
Page 1, Line 4-6: "Results showed that antimicrobial-resistant genes and antibiotic-resistant bacteria in organisms cause critical infectious diseases." This does not make sense. rephrase. Also, change "antimicrobial-resistant genes" to " antimicrobial resistance genes"
Authors’ Response 9
They were fixed.

Round 2
Reviewer 1 Report
Comments and Suggestions for Authors
This manuscript has been revised. All necessary modifcations have been done in the text.
Reviewer 2 Report
Comments and Suggestions for Authors
Authors have answered all comments. I accept this manuscript in current status